# Seasonal Influenza Vaccine Intention among Nurses Who Have Been Fully Vaccinated against COVID-19: Evidence from Greece

**DOI:** 10.3390/vaccines11010159

**Published:** 2023-01-11

**Authors:** Petros Galanis, Aglaia Katsiroumpa, Irene Vraka, Olga Siskou, Olympia Konstantakopoulou, Theodoros Katsoulas, Daphne Kaitelidou

**Affiliations:** 1Clinical Epidemiology Laboratory, Faculty of Nursing, National and Kapodistrian University of Athens, 11527 Athens, Greece; 2Department of Radiology, P. & A. Kyriakou Children’s Hospital, 11527 Athens, Greece; 3Department of Tourism Studies, University of Piraeus, 18534 Piraeus, Greece; 4Center for Health Services Management and Evaluation, Faculty of Nursing, National and Kapodistrian University of Athens, 11527 Athens, Greece; 5Faculty of Nursing, National and Kapodistrian University of Athens, 11527 Athens, Greece

**Keywords:** influenza, vaccine, intention, nurses, COVID-19

## Abstract

Seasonal influenza continues to be a significant public health issue causing hundreds of thousands of deaths annually. Nurses are a priority risk group for influenza vaccination and a high vaccine uptake rate among them is crucial to protect public health. Thus, the aim of our study was to estimate the levels of influenza vaccine acceptance, as well as its determinants, among nurses for the 2022/2023 season. We conducted a cross-sectional study with a convenience sample in Greece. We collected data via an online survey in September 2022. Most of the nurses in the study possessed a MSc/PhD diploma (56.2%) and had previously been infected by SARS-CoV-2 (70.3%). Among nurses, 57.3% were willing to accept the influenza vaccine, 19% were hesitant, and 23.7% were unwilling. Older age, higher levels of perceived support from significant others, and higher COVID-19-related physical exhaustion were positively related to influenza vaccination intention. In contrast, more side effects because of COVID-19 vaccination and higher levels of exhaustion due to measures taken against COVID-19 were negatively associated with vaccination intention. Since the influenza vaccination acceptance rate among nurses was moderate, policymakers should develop and implement measures tailored specifically to nurses in the context of the COVID-19 pandemic to decrease vaccine hesitancy.

## 1. Introduction

Seasonal influenza is a significant health threat since it causes hundreds of thousands of deaths, substantial economic burden, and considerable social consequences [1,2]. For instance, influenza infects approximately 50 million individuals in the European Union/European Economic Area and 15,000–70,000 European citizens die from influenza-related respiratory diseases each year [3]. Globally, influenza affects 5–15% of the population, leading to about 650,000 deaths on an annual basis [4].

Evidence suggests that influenza vaccination can protect against influenza and influenza-like illnesses in healthy adults, healthy children under 16 years, the elderly, and patients such as those with chronic obstructive pulmonary disease [5,6,7,8]. Additionally, systematic reviews have identified that influenza vaccination is a cost-saving or cost-effective intervention [9,10]. Moreover, a recent meta-analysis [11] showed that influenza vaccination was associated with a reduced risk of mechanical ventilation among COVID-19 patients, while another meta-analysis [12] showed a lower risk of COVID-19 infection and influenza leading to hospitalization among the vaccinated group compared to the non-vaccinated group.

Based on the existing evidence, the Centers for Disease Control and Prevention (CDC) recommends annual influenza vaccination for all persons aged ≥ 6 months with few exceptions [13]. Moreover, this recommendation is given greater emphasis for healthcare workers in healthcare settings who are most at risk of infection, in order to protect themselves and their vulnerable patients from influenza and to ensure the overall safe running of services [14,15]. 

Increased coverage of influenza vaccination among healthcare workers during the ongoing COVID-19 pandemic is crucial to mitigate the transmission of the influenza virus. Moreover, the influenza prevalence is higher among healthcare workers (6.3%) compared to healthy working adults (5.1%) [16,17], while healthcare workers experience significant morbidity and mortality due to occupational exposure to the influenza virus [17,18]. The need for influenza vaccination is particularly high for nurses since they are frontline healthcare workers who spend far more time with patients than other workers in healthcare settings [19]. The influenza vaccination uptake rate among nurses in England was higher for 2021–2022 (76.3% for nurses in hospitals and 81.5% for nurses in primary healthcare settings) compared with 2020–2021 (61.5% for nurses in hospitals and 76.9% for nurses in primary healthcare settings) [20]. In contrast, a slight decrease in the influenza vaccination uptake rate among nurses in the USA was reported by the CDC (87.8% for 2021–2022 compared to 90.3% for 2020–2021, and 92% for 2019–2020) [21].

Nurses’ attitudes towards influenza vaccination for 2022–2023 are important since the past two seasons have seen historically low levels of influenza and an increase in influenza activity is expected. Before this study, only Sallam et al. [22] had measured the intentions of healthcare workers, including nurses, to accept influenza vaccination (for 2021–2022), and they found that the acceptance rates were highest among males, physicians, dentists, and healthcare workers with high confidence in the vaccine and a strong sense that it was part of their collective responsibility to get vaccinated. To test whether the same would be found, we estimated the levels of influenza vaccine acceptance, as well as its determinants, among nurses in Greece for the2022/2023 season.

## 2. Materials and Methods

### 2.1. Study Design and Sample

We conducted a cross-sectional study in Greece where data were collected via an online survey in September 2022. The influenza vaccine was provided free of cost for all citizens in Greece from October 2022. Therefore, we measured nurses’ intention to accept the influenza vaccine just before the rollout of the vaccine. We constructed the online questionnaire with Google Forms, and we shared it through social media. Additionally, we sent the questionnaire to nurses in our email contacts and we asked them to share the questionnaire link with other nurses. Thus, we applied a snowball sampling technique. Accordingly, a convenience sample was obtained. Nurses who understood the Greek language and worked in clinical settings could participate in our study. Considering a low effect size (f^2^ = 0.02), the precision level of 5% (alpha level), the power of 95%, and the number of predictors as 13, a minimum sample size of 652 nurses was required. We decided to recruit more nurses in order to further decrease random error in our study.

### 2.2. Measures

The online questionnaire comprised four sections: (a) general characteristics of nurses, (b) influenza vaccination intention, (c) perceived social support, and (d) COVID-19-related burnout. 

General characteristics of nurses included gender (females or males), age (continuous variable), MSc/PhD diploma (no or yes), chronic disease (no or yes), clinical experience (continuous variable), SARS-CoV-2 infection (no or yes), and side effects because of COVID-19 vaccination (continuous variable on a scale from 0 (no side effects) to 10 (too many side effects)).

The outcome variable was the nurses’ intention to accept the influenza vaccine. We used the following question to measure this intention: “How likely do you think you are to get the influenza vaccine”? Answers were on a scale from 0 (very unlikely) to 10 (very likely). We used a priori cut-off points in order to discriminate between willing, hesitant, and unwilling nurses. Nurses with a score from 0 to 2 were considered unwilling to accept the influenza vaccine, nurses with a score from 3 to 7 as hesitant, and nurses with a score from 8 to 10 as willing. 

We measured the social support that nurses received with the Multidimensional Scale of Perceived Social Support (MSPSS) [23]. The MSPSS includes 12 items with answers on a 7-point Likert scale (1: strongly disagree; 2: disagree; 3: disagree to some extent; 4: neutral; 5: agree to some extent; 6: agree; 7: strongly agree). The MSPSS assesses three factors: support from family, support from friends, and support from significant others. All factors take values from 1 (low social support) to 7 (high social support). The MSPSS has been validated in Greek [24]. In our study, Cronbach’s alpha for support from family was equal to 0.95, for support from friends was equal to 0.96, and for support from significant others was equal to 0.92. 

We used the COVID-19 burnout scale (COVID-19-BS) to measure COVID-19-related burnout [25]. The COVID-19-BS comprises 13 items and answers are on a 5-point Likert scale (1: strongly disagree; 2: disagree; 3: neutral; 4: agree; 5: strongly agree). The COVID-19-BS consists of three factors: COVID-19-related emotional exhaustion, COVID-19-related physical exhaustion, and exhaustion due to measures taken against COVID-19 (e.g., wearing a face mask, rapid tests, PCR tests, isolation, and vaccination). The overall score on each factor ranges from 1 (low level of burnout) to 5 (high level of burnout). In our study, Cronbach’s alpha for emotional exhaustion was equal to 0.88, for physical exhaustion was equal to 0.87, and for exhaustion due to measures against COVID-19 was equal to 0.88.

### 2.3. Ethical Considerations

We did not collect personal data from nurses. Participation was anonymous and voluntary. Additionally, we informed nurses about the aim and the design of our study. Moreover, we ensured they gave electronic informed consent by asking, “Do you agree to participate in this study”? as the first question on the online questionnaire. Nurses with positive answers could then participate in our study. 

We conducted our study according to the guidelines of the Declaration of Helsinki. The Ethics Committee of the Faculty of Nursing, National and Kapodistrian University of Athens approved the study protocol (reference number: 370, 2 September 2021).

### 2.4. Statistical Analysis

We used the mean, standard deviation, median, minimum value, and maximum value to present continuous variables. Additionally, we used numbers and percentages to express categorical variables. The influenza vaccination intention score followed a normal distribution. Therefore, we used linear regression analysis to investigate the impact of independent variables on vaccination intention. We considered demographic characteristics, job characteristics, COVID-19-related variables, social support, and COVID-19-related burnout as independent variables. First, we performed univariate linear regression analysis, and then we constructed a final multivariable model, simultaneously taking into consideration all the independent variables. We present unadjusted and adjusted beta coefficients, 95% confidence intervals (CI), and *p*-values. Moreover, we performed a sensitivity analysis to investigate the robustness of the final multivariable linear regression model. To do so, we performed univariate and multivariable logistic regression analysis. We considered nurses with scores ≥ 8 on the influenza vaccination intention scale as willing to accept the influenza vaccine and nurses with scores ≤ 7 as hesitant/unwilling. We present unadjusted and adjusted odds ratios, 95% CIs, and *p*-values. The significance level was set to 0.05 in all cases. Statistical analysis was performed with the Statistical Package for Social Sciences software (IBM SPSS Statistics for Windows, Version 21.0., IBM Corp., Armonk, NY, USA).

## 3. Results

The final sample included 861 nurses, with a predominance of females (87.9%) and a mean age of 37.9 years (minimum: 22, maximum: 63). Interestingly, most nurses possessed a MSc/PhD diploma (56.2%). The mean number of years of clinical experience was 12.2 (minimum: 1, maximum: 36). The majority of nurses (70.3%) had been infected by SARS-CoV-2 during the pandemic. Most nurses (87.5%) experienced at least minor side effects because of COVID-19 vaccination. The general characteristics of nurses are shown in Table 1.

Among nurses, 57.3% (n = 493) were willing to accept the influenza vaccine, 19% (n = 164) were hesitant, and 23.7% (n = 204) were unwilling. Additionally, the mean influenza vaccination intention score was 6.69, indicating a moderate-to-high level of acceptance (Table 2). 

Focusing on social support, we found that nurses perceived high levels of support. Specifically, nurses perceived the most support from significant others (mean score: 6.03), the second most from family (mean score: 5.89), and the least from friends (mean score: 5.70) (Table 2). 

Nurses were found to experience moderate levels of burnout related to COVID-19. Specifically, nurses experienced the most burnout due to measures against COVID-19 (mean score: 3.48), followed by emotional burnout (mean score: 3.43), and physical burnout (mean score: 2.69) (Table 2).

Table 3 shows the result of a univariate and multivariable linear regression analysis of factors affecting the influenza vaccination intention of nurses. Older age (adjusted beta: 0.08, 95% CI: 0.01 to 0.15), higher levels of perceived support from significant others (adjusted beta: 0.39, 95% CI: 0.09 to 0.68), and higher COVID-19-related physical exhaustion (adjusted beta: 0.42, 95% CI: 0.09 to 0.75) were positively related to influenza vaccination intention. In contrast, more side effects because of COVID-19 vaccination (adjusted beta: −0.22, 95% CI: −0.32 to −0.11), and higher levels of exhaustion due to measures against COVID-19 (adjusted beta: −0.74, 95% CI: −0.99 to −0.49) were negatively associated with vaccination intention, so nurses were less likely to accept the influenza vaccine. Sensitivity analysis using logistic regression models confirmed the results of the linear regression analysis (Table 4). Therefore, in the final multivariable logistic regression model, intention to get the influenza vaccine was associated with being older, perceiving more support from significant others, experiencing more COVID-19-related physical exhaustion, experiencing less exhaustion due to measures against COVID-19, and experiencing fewer side effects because of COVID-19 vaccination.

## 4. Discussion

To the best of our knowledge, this is the first study on the intention of nurses to accept the influenza vaccine for the2022/2023 season. Additionally, we investigated potential determinants of nurses’ willingness to get the influenza vaccine.

In our study, the vaccination acceptance rate was 57.3%, while 23.7% of nurses were unwilling to get the influenza vaccine and 19% were hesitant. An earlier study in Greece [26] found that 50.3% of nurses got the influenza vaccine for the 2021/2022 season. Additionally, Sallam et al. [22] found that 64.6% of a sample of nurses in Jordan were willing to accept the influenza vaccine for the 2021/2022 season. In studies in China [27] and Turkey [28], the influenza vaccine uptake prevalence among nurses for the 2021/2022 season was 49% and 10.3%, respectively. However, the influenza vaccine uptake rate for the same season was much higher among nurses in the USA and England (87.8% and 78.9%, respectively) [20,21]. These great differences in seasonal influenza vaccination coverage among countries could be attributed to contextual, cultural, psychological, and sociodemographic factors [29]. The unwillingness of nurses in our study to get the influenza vaccine is worrying, and the actual uptake rate could potentially be lower than the intention rate. To improve the situation, policymakers should develop and implement combined educational interventions and training programs tailored to individual healthcare settings in order to improve nurses’ knowledge about the influenza vaccine and increase vaccine uptake [30]. Evidence-based approaches and the development of decision-making skills could empower nurses to overcome their vaccine hesitancy and thus improve vaccine uptake rates.

In our study, nurses who perceived that they had greater social support from significant others were more likely to get the influenza vaccine than nurses who perceived having less social support. It appears that social support could act as an external resource that empowers nurses to adopt positive attitudes such as vaccine uptake. Additionally, studies during the COVID-19 pandemic suggested the protective role of social support from significant others (e.g., co-workers and managers) in safeguarding against mental health consequences among nurses [31,32]. Moreover, the literature contends that social support may have acted not only directly but also indirectly as a mediating variable by reducing anxiety, insomnia symptoms, and turnover intention among nurses during the pandemic [33,34,35]. 

Our results showed that several COVID-19-related variables were linked to nurses’ willingness to accept the influenza vaccine. In particular, nurses who experienced more side effects because of COVID-19 vaccination were less likely to take the influenza vaccine for the 2022/2023 season. In addition, we found that increased exhaustion due to measures taken against COVID-19, including vaccination, was associated with reduced influenza vaccination intention among nurses. Thus, COVID-19 vaccination seems to have a negative impact on nurses’ intentions to accept the influenza vaccine either directly through side effects or indirectly through vaccination exhaustion. It is well-known that nurses’ fears of influenza vaccine side effects negatively affect their likelihood of accepting it [36,37]. Additionally, nurses’ decision to accept influenza vaccination is based on sources of information [38] and their perception of being surrounded by an (un)trustworthy environment [39]. Lack of knowledge and misconceptions about influenza and influenza vaccines are also significant determinants of nurses’ hesitancy [37]. For instance, increased hesitancy is caused by ill-founded immunological beliefs among nurses, such as the belief that hand washing, wearing face masks, and personal immunity are more effective than the influenza vaccine [36]. Moreover, nurses view influenza vaccination mainly as a personal decision and not as an evidence-based nursing measure [38,40]. Thus, personal motivators are strong predictors of healthcare workers’ vaccination uptake and their decisions are made in the context of personal health choices [41]. A review found that the self-protection of healthcare workers is the most important reason for influenza vaccination acceptance [37]. Thus, policymakers should develop and implement interventions tailored specifically to nurses in order to improve the nurses’ knowledge, increase their confidence in vaccines, and decrease their vaccine hesitancy. A lack of knowledge about influenza vaccines and influenza could be improved by future educational programs that address concerns about vaccine safety, enhance awareness and understanding of the vaccine, emphasize personal vaccination benefits, and highlight the negative impact of nurses’ hesitancy to get vaccinated on patients and public health. There is evidence to suggest such a program may be effective; for example, a randomized controlled trial among healthcare workers in nursing homes showed that influenza vaccination uptake was 25% in the intervention group that received multi-faceted education and institutional support compared to 16% in the control group [42].

According to our results, increased COVID-19-related physical exhaustion among nurses was related to increased influenza vaccination intention. The COVID-19 pandemic increased the physical and psychological strain on nurses, leading to several mental health issues, such as depression, anxiety, post-traumatic stress disorder, stress, sleep disorders, and burnout [43,44,45,46]. Moreover, nurses experienced moderate-to-high levels of fear during the pandemic [47,48,49,50]. It is probable that three years after the onset of the pandemic, exhausted nurses are more afraid than they were pre-pandemic of getting an influenza infection in the coming winter season. Yet, several studies suggest that COVID-19 fear is positively associated with acceptance of and adherence to preventive measures among healthcare workers and the general public [51,52,53]. It seems to be that the higher the perceived susceptibility, the more precautions are taken by the individuals. 

Additionally, we found that older nurses were more likely to accept the influenza vaccine than younger nurses. Several studies suggest that older healthcare workers are significantly associated with higher influenza vaccine uptake and willingness compared to younger age groups, even during the COVID-19 pandemic [22,54,55,56,57,58]. This could be explained by old age being an independent predictor of severe complications from influenza virus infections such as pneumonia [59,60]. Moreover, older influenza-positive patients are at higher risk of hospitalization than younger patients [61]. Thus, the high influenza vaccine intention among older nurses may be attributed to their awareness of being members of a risk group. It is probable that older nurses feel more vulnerable than younger ones and consider influenza as a potentially serious disease.

### Limitations

Our study had several limitations. First, we used a convenience sample with online recruitment and, therefore, selection bias was potentially a factor. Studies with nationally representative samples could add invaluable information. Second, nurses’ self-reporting of influenza vaccine intention could have added information bias to our study. In addition, the voluntary nature of participation in our study may have increased the participation of nurses with positive attitudes toward vaccination. Data from registry records on actual vaccine uptake among nurses in Greece could overcome this bias. Third, we investigated several potential determinants of nurses’ willingness to accept the influenza vaccine but there is still room to assess the impact of other predictors, e.g., psychological predictors. Fourth, we conducted a cross-sectional study, in which we could not infer causal relationships between independent variables and vaccine intention. Longitudinal studies are necessary to capture nurses’ attitudes toward vaccination over time. Fifth, we did not measure the exact side effects of COVID-19 vaccination since our study questionnaire included a great number of variables. Further studies could measure the side effects of COVID-19 vaccination with greater validity. Finally, we collected our data through social media and our email contacts using the snowball sampling technique. We could not assess the numbers and characteristics of the non-respondents since we did not know the nurses who saw our questionnaire as a Google Form but did not answer it. However, to obtain robust results, we recruited more participants than the minimum sample size. This helped us reduce random error, and logistic regression analysis produced narrow confidence intervals for the odds ratios. By adding extra nurses to our study, we produced more reliable estimations for measures of association than if we had stuck to the minimum sample size.

## 5. Conclusions

Our findings offer important clues on how to reduce influenza vaccine hesitancy among nurses. We should note that our sample included mainly highly educated nurses who have been infected by SARS-CoV-2 during the pandemic. Understanding the factors that affect nurses’ decisions to accept the influenza vaccine is highly recommended in order to reduce physical and psychological barriers and to improve actual vaccine uptake. As evidenced by our findings, COVID-19-related variables, such as vaccination fatigue and side effects from previous COVID-19 vaccination, negatively affect nurses’ willingness to accept the influenza vaccine. Thus, policymakers should develop and implement a multi-faceted set of interventional measures in the context of the COVID-19 pandemic. These measures should be tailored specifically to nurses and could include educational programs, psychological support, promotion of decision-making skills, and free on-site vaccination. In addition, policymakers should be aware that the degree of adoption of the influenza vaccine in the elderly is associated with greater social responsibility and less “internet hysteria” than are noted in younger age groups. Nurses are a priority risk group for influenza vaccination and a high vaccine uptake rate among them is crucial to protect their family members, their patients, and their community. Political and scientific authorities should recognize that the level of influenza vaccination acceptance among nurses is moderate and efforts are required to increase it. To that end, the identification of factors related to nurses’ intention to be vaccinated is crucial to improve the vaccination rate. We found that the influenza vaccination acceptance rates were particularly low among specific groups, i.e., younger nurses, nurses who perceived having low support, nurses who experienced many side effects of COVID-19 vaccination, and nurses with high levels of exhaustion due to measures against COVID-19. Therefore, political and scientific authorities should target these groups when implementing appropriate interventions, such as those offering up-to-date and evidence-based information on the safety, effectiveness, and efficacy of COVID-19 vaccines, reduced measures against COVID-19, and psychological support from experienced healthcare professionals. Three years after the onset of the COVID-19 pandemic, nurses are experiencing high levels of burnout. Support from political and scientific authorities is crucial to improve nurses’ spirit and self-confidence and to increase their influenza vaccination rate.

## Figures and Tables

**Table 1 vaccines-11-00159-t001:** General characteristics of nurses (N = 861).

Characteristics	N	%
Gender		
Males	104	12.1
Females	757	87.9
Age (years)	37.9 ^a^	9.8 ^b^
MSc/PhD diploma		
No	377	43.8
Yes	484	56.2
Chronic disease		
No	642	74.6
Yes	219	25.4
Clinical experience (years)	12.2 ^a^	9.3 ^b^
SARS-CoV-2 infection		
No	256	29.7
Yes	605	70.3
Side effects because of COVID-19 vaccination	840	97.5

^a^ mean. ^b^ standard deviation.

**Table 2 vaccines-11-00159-t002:** Descriptive statistics for the scales in this study.

Scale	Mean	Standard Deviation	Median	Minimum Value	Maximum Value
Influenza vaccination intention	6.69	3.85	9	0	10
Support from significant others	6.03	1.35	6.5	1	7
Support from family	5.89	1.48	6.5	1	7
Support from friends	5.70	1.46	6	1	7
COVID-19-related emotional exhaustion	3.43	1.10	3.4	1	5
COVID-19-related physical exhaustion	2.69	1.14	2.5	1	5
Exhaustion due to measures against COVID-19	3.48	1.26	3.5	1	5

**Table 3 vaccines-11-00159-t003:** Univariate and multivariable linear regression analysis with influenza vaccination intention score as the dependent variable.

Variable	Unadjusted Beta Coefficient (95% CI)	*p*-Value	Adjusted Beta Coefficient (95% CI) ^a^	*p*-Value
Gender (females vs. males)	−1.04 (−1.82 to −0.25)	0.01	−0.46 (−1.31 to 0.38)	0.28
Age (years)	0.07 (0.05 to 0.10)	<0.001	0.08 (0.01 to 0.15)	**0.02**
MSc/PhD diploma (yes vs. no)	0.32 (−0.20 to 0.84)	0.22	0.13 (−0.41 to 0.67)	0.63
Chronic disease (yes vs. no)	0.69 (0.09 to 1.28)	0.02	0.37 (−0.25 to 0.99)	0.24
Clinical experience (years)	0.07 (0.04 to 0.09)	<0.001	−0.01 (−0.08 to 0.06)	0.77
SARS-CoV-2 infection (yes vs. no)	−0.03 (−0.59 to 0.54)	0.93	0.20 (−0.38 to 0.79)	0.49
Side effects because of COVID-19 vaccination	−0.26 (−0.37 to −0.16)	<0.001	−0.22 (−0.32 to −0.11)	**<0.001**
Support from significant others	0.39 (0.21 to 0.58)	<0.001	0.39 (0.09 to 0.68)	**0.01**
Support from family	0.27 (0.09 to 0.44)	0.003	−0.002 (−0.25 to 0.24)	0.99
Support from friends	0.23 (0.05 to 0.41)	0.01	0.11 (−0.13 to 0.36)	0.38
COVID-19-related emotional exhaustion	−0.12 (−0.36 to 0.11)	0.31	0.24 (−0.12 to 0.60)	0.19
COVID-19-related physical exhaustion	0.03 (−0.20 to 0.25)	0.83	0.42 (0.09 to 0.75)	**0.01**
Exhaustion due to measures against COVID-19	−0.79 (−0.99 to −0.60)	<0.001	−0.74 (−0.99 to −0.49)	**<0.001**

Bold *p*-values indicate statistically significant associations in the multivariable model. CI: confidence interval. ^a^ R^2^ for the final multivariable model was 13.2%.

**Table 4 vaccines-11-00159-t004:** Univariate and multivariable logistic regression analysis with influenza vaccination willingness as the dependent variable (reference category: hesitant/unwilling nurses).

Variable	Unadjusted OR (95% CI)	*p*-Value	Adjusted OR (95% CI) ^a^	*p*-Value
Gender (females vs. males)	0.45 (0.29 to 0.71)	0.001	0.60 (0.35 to 1.03)	0.06
Age (years)	1.05 (1.03 to 1.06)	0.001	1.05 (1.01 to 1.09)	**0.02**
MSc/PhD diploma (yes vs. no)	1.14 (0.87 to 1.50)	0.34	0.98 (0.71 to 1.35)	0.90
Chronic disease (yes vs. no)	1.49 (1.08 to 2.05)	0.01	1.23 (0.84 to 1.80)	0.28
Clinical experience (years)	1.05 (1.03 to 1.07)	<0.001	1.00 (0.96 to 1.05)	0.87
COVID-19 diagnosis (yes vs. no)	0.88 (0.66 to 1.19)	0.41	1.08 (0.76 to 1.54)	0.68
Side effects because of COVID-19 vaccination	0.87 (0.82 to 0.92)	<0.001	0.87 (0.82 to 0.93)	**<0.001**
Support from significant others	1.18 (1.07 to 1.31)	0.001	1.31 (1.09 to 1.56)	**0.004**
Support from family	1.12 (1.02 to 1.22)	0.02	0.99 (0.86 to 1.16)	0.99
Support from friends	1.06 (0.97 to 1.16)	0.22	0.98 (0.84 to 1.14)	0.79
COVID-19-related emotional exhaustion	0.96 (0.85 to 1.08)	0.48	1.20 (0.96 to 1.49)	0.11
COVID-19-related physical exhaustion	1.02 (0.91 to 1.15)	0.70	1.23 (1.01 to 1.50)	**0.04**
Exhaustion due to measures against COVID-19	0.66 (0.59 to 0.75)	<0.001	0.67 (0.57 to 0.79)	**<0.001**

Bold *p*-values indicate statistically significant associations in the multivariable model. CI: confidence interval; OR: odds ratio. ^a^ R^2^ for the final multivariable model was 20%.

## Data Availability

The data presented in this study are available on request from the corresponding author.

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
