# Peer review of "Seasonal Influenza Vaccine Intention among Nurses Who Have Been Fully Vaccinated against COVID-19: Evidence from Greece"

_vaccines, 2023, doi:10.3390/vaccines11010159_

Round 1

Reviewer 1 Report

The work makes a good impression.

The work is sociological in nature. I believe that the article will be interesting and useful to a wide range of infectionists. The authors received, in my opinion, a fairly obvious result, but they confirmed it with their study with a completely adequate mathematical method.

As for the authors' conclusions, it can be added to them that the degree of adoption of the influenza vaccine in the elderly is associated with greater social responsibility and less adherence to "Internet hysteria".

Author Response

Dear Reviewer,

Thank you for giving us the opportunity to revise our manuscript entitled "Seasonal influenza vaccine intention among nurses who have been fully vaccinated against COVID-19: Evidence from Greece". We would also like to thank you for your insightful comments and suggestions on how to improve our manuscript. We respectfully tried to address the issues raised and to revise our manuscript accordingly. We hope that our revision will reach the high standards of the journal “Vaccines”.

We are grateful for your comments. You really help us to improve our manuscript. We apply all your suggestions in our manuscript.

Also, we made changes in the manuscript according to the other Reviewers’ instructions.

We look forward to hearing from you

Best Regards

The authors

Comments from Reviewer

As for the authors' conclusions, it can be added to them that the degree of adoption of the influenza vaccine in the elderly is associated with greater social responsibility and less adherence to "Internet hysteria".

Answer: Done

Dear Reviewer, please see the Conclusions section. We add the following:

In addition, policy makers should have in their minds that the degree of adoption of the influenza vaccine in the elderly is associated with greater social responsibility and less adherence to “internet hysteria”.

Reviewer 2 Report

Methods

This is a straightforward survey of willingness and intention to obtain influenza vaccination, using email contacts and snowballing.

It is important in the abstract, results and conclusions sections to stress that your sample is highly skewed to nurses with an MSc./Ph/D (56%) and who had had SARS-CoV-2 (70.3%).

Please describe the numbers and characteristics of the non-respondents and how you added more participants and the changes in your results from adding more participants. 

Results

The authors found that four variables were statistically significantly related to intention to be vaccinated: age, side effects, exhaustion and significant others support. 

Please state any recommendations you have for political and medical authorities whose responsbibilty it is is to increase vaccination rates.

Author Response

Dear Reviewer,

Thank you for giving us the opportunity to revise our manuscript entitled "Seasonal influenza vaccine intention among nurses who have been fully vaccinated against COVID-19: Evidence from Greece". We would also like to thank you for your insightful comments and suggestions on how to improve our manuscript. We respectfully tried to address the issues raised and revise our manuscript accordingly. We hope that our revision will reach the high standards of the journal “Vaccines”.

We are grateful for your comments. You really help us to improve our manuscript. We apply all your suggestions to our manuscript.

Also, we made changes in the manuscript according to the other Reviewers’ instructions.

We look forward to hearing from you

Best Regards

The authors

Comments from Reviewer

It is important in the abstract, results and conclusions sections to stress that your sample is highly skewed to nurses with an MSc./Ph/D (56%) and who had had SARS-CoV-2 (70.3%).

Answer: Done

Dear Reviewer, please see the abstract, results and conclusions sections. We remove another sentence from the Abstract in order to reach the limit number of 200 words in the Abstract section.

Please describe the numbers and characteristics of the non-respondents and how you added more participants and the changes in your results from adding more participants. 

Answer: Done

Dear Reviewer, thank you a lot for this comment since it is a limitation that we have not recognized in our limitations. Now, we add this limitation to our manuscript. Please, see the Limitations section. We add the following:

“Finally, we collected our data through social media and our email contacts using the snowball sampling technique. Thus, we cannot assess the numbers and characteristics of the non-respondents since we did not know the nurses that saw our questionnaire as a Google form but they did not answer it.”

Dear Reviewer, unfortunately, we cannot describe the numbers and characteristics of the non-respondents since we collected our data through social media and our email contacts using the snowball sampling technique. Thus, we do not know the numbers and characteristics of nurses that see our questionnaire as a Google form but they did not answer it.

Also, we describe the changes in our results from adding more participants in the Limitations section. We add the following:

“However, in order to get more robust results we recruited more participants than the minimum sample size. In that way, we reduced random error and logistic regression analysis produced more narrow confidence intervals for odds ratios. Thus, by adding more nurses to our study we produced more reliable estimations for measures of association.”

Results

The authors found that four variables were statistically significantly related to intention to be vaccinated: age, side effects, exhaustion and significant others support. 

Please state any recommendations you have for political and medical authorities whose responsibility it is to increase vaccination rates.

Answer: Done

Dear Reviewer, please see the Conclusions section. We add the following:

“Thus, political and scientific authorities should recognize that the level of influenza vaccination acceptance rate among nurses is moderate and make efforts to increase it. Moreover, the identification of factors that are related to nurses’ intention to be vaccinated is crucial to improve the vaccination rate. We found that influenza vaccination acceptance rate is lower among specific groups, i.e. younger nurses, nurses who perceived less support, and nurses who experienced more side effects because of COVID-19 vaccination and higher levels of exhaustion due to measures against the COVID-19. Therefore, political and scientific authorities should pay more attention to these groups implementing the appropriate interventions, such as update and evidence based knowledge regarding the safety, effectiveness, and efficacy of COVID-19 vaccines, reduced measures against the COVID-19, and psychological support from experienced healthcare professionals. Three years after the onset of the COVID-19 pandemic nurses experience high levels of burnout and support from political and scientific authorities is pivotal to improve nurses’ spirit and self-confidence and increase influenza vaccination rate.”

Reviewer 3 Report

In this study, the authors analyzed the levels of influenza vaccine acceptance as well as its determinants among nurses. They conducted a cross-sectional study with a convenience sample in Greece, collected data via an online survey during September 2022, and measured socio-demo-graphic data of nurses, influenza vaccination intention, perceived social support, and COVID-19-related burnout. The conclusion can provide clues to reduce the influenza vaccine hesitancy among nurses, which is crucial to protect their family members, their patients, and their community. Overall, the study is well-designed and the manuscript is clear. I have some minor comments for improvement.

1. “However, influenza vaccination uptake rate among nurses in England was lower for 2021 to 2022 (76.3% for nurses in hospitals and 81.5% for nurses in primary healthcare settings) compared with 2020 to 2021 (61.5% for nurses in hospitals and 76.9% for nurses in primary healthcare settings)”. The number of 2021-2022 was higher than that of 2020-2021.

2. “We used a priori cut-off points in order to discriminate between willing, hesitant, and unwilling nurses.”. What is the “priori cut-off points”?

3. “Most nurses (87.5%) experienced even minor side effects because of COVID-19 vaccination. General characteristics of nurses are shown in Table 1.” The percentage should be 97.5% based on Table 1.

4. No standard deviation data in Table 1.

5. What are the side effects of COVID-19 vaccination? Why not include COVID-19-related exhaustion? Why side effects and COVID-19-related exhaustion have different effect on influenza vaccination willingness?

6. It should be “the Centers for Disease Control and Prevention (CDC)”

Author Response

Dear Reviewer,

Thank you for giving us the opportunity to revise our manuscript entitled "Seasonal influenza vaccine intention among nurses who have been fully vaccinated against COVID-19: Evidence from Greece". We would also like to thank you for your insightful comments and suggestions on how to improve our manuscript. We respectfully tried to address the issues raised and revise our manuscript accordingly. We hope that our revision will reach the high standards of the journal “Vaccines”.

We are grateful for your comments. You really help us to improve our manuscript. We apply all your suggestions to our manuscript.

Also, we made changes in the manuscript according to the other Reviewers’ instructions.

We look forward to hearing from you.

Best Regards,

The authors

Comments from Reviewer

In this study, the authors analyzed the levels of influenza vaccine acceptance as well as its determinants among nurses. They conducted a cross-sectional study with a convenience sample in Greece, collected data via an online survey during September 2022, and measured socio-demo-graphic data of nurses, influenza vaccination intention, perceived social support, and COVID-19-related burnout. The conclusion can provide clues to reduce the influenza vaccine hesitancy among nurses, which is crucial to protect their family members, their patients, and their community. Overall, the study is well-designed and the manuscript is clear. I have some minor comments for improvement.

  1. “However, influenza vaccination uptake rate among nurses in England was lower for 2021 to 2022 (76.3% for nurses in hospitals and 81.5% for nurses in primary healthcare settings) compared with 2020 to 2021 (61.5% for nurses in hospitals and 76.9% for nurses in primary healthcare settings)”. The number of 2021-2022 was higher than that of 2020-2021.

Answer: Done

Dear Reviewer, thank you a lot for your sharp eye.

  1. “We used a priori cut-off points in order to discriminate between willing, hesitant, and unwilling nurses.”. What is the “priori cut-off points”?

Answer: Done

Dear Reviewer, we used a scale from 0 (very unlikely) to 10 (very likely) to measure the intention of nurses to accept the influenza vaccine. In order to categorize nurses as willing, hesitant, and unwilling nurses to accept the influenza vaccine, we considered a priori cut-off points before the data collection. A priori cut-off points do not take into account the statistical properties of the variable after the data collection, e.g. normal distribution or not. We used a priori cut-off points in order to avoid possible bias that could statistics introduce. Thus, we considered a priori that nurses with a score from 0 to 2 were unwilling to accept the influenza vaccine, nurses with a score from 3 to 7 were hesitant, and nurses with a score from 8 to 10 were willing.

Do you want to add more details in the Methods section regarding the “priori cut-off points”?

  1. “Most nurses (87.5%) experienced even minor side effects because of COVID-19 vaccination. General characteristics of nurses are shown in Table 1.” The percentage should be 97.5% based on Table 1.

Answer: Done

Dear Reviewer, thank you again for your sharp eye. Please, see Table 1.

  1. No standard deviation data in Table 1.

Answer: Done

Dear Reviewer, we fix it. Please, see Table 1.

  1. What are the side effects of COVID-19 vaccination? Why not include COVID-19-related exhaustion? Why side effects and COVID-19-related exhaustion have different effect on influenza vaccination willingness?

Answer: Done

Dear Reviewer, we did not measure the exact side effects of COVID-19 vaccination since the length of the study questionnaire was long including many variables. Thank you a lot for this comment since it is a limitation that we have not recognized in our limitations. Now, we add this limitation to our manuscript. Please, see the Limitations section. We add the following:

“Fifth, we did not measure the exact side effects of COVID-19 vaccination since our study questionnaire included a great number of variables. Further studies could measure side effects of COVID-19 vaccination in a more valid way.”

We used a reliable and valid tool (COVID-19 burnout scale) to measure COVID-19-related burnout. The COVID-19 burnout scale consists of three factors: COVID-19-related emotional exhaustion, COVID-19-related physical exhaustion, and exhaustion due to measures against the COVID-19. Thus, the COVID-19 burnout scale does not measure the side effects of COVID-19 vaccination. This is the reason that we added a question regarding the side effects of COVID-19 vaccination in our study questionnaire. We think that is also the reason why side effects and COVID-19-related exhaustion have a different effect on influenza vaccination willingness. Side effects and COVID-19-related exhaustion are two different concepts and may act in different ways. Side effects act negatively since nurses’ fear of influenza vaccine side effects affect negatively their decision to accept it. On the other hand, nurses’ COVID-19-related physical exhaustion acts positively. It is probable that three years after the onset of the pandemic, exhausted nurses are more afraid of influenza infection in the next winter season. We present all this information in the fourth and fifth paragraphs in the Discussion section. Do you want to add something more?

  1. It should be “the Centersfor Disease Control and Prevention (CDC)”

Answer: Done

We correct it.